# Lichen Amyloidosis in an Atopic Patient Treated with Dupilumab: A New Therapeutic Option

**DOI:** 10.3390/diseases12050094

**Published:** 2024-05-06

**Authors:** Benedetta Tirone, Gerardo Cazzato, Francesca Ambrogio, Caterina Foti, Marco Bellino

**Affiliations:** 1Section of Dermatology and Venereology, Department of Precision and Regenerative Medicine and Ionian Area (DiMePRe-J), University of Bari “Aldo Moro”, 70124 Bari, Italy; benedetta.ti96@gmail.com (B.T.); dottambrogiofrancesca@gmail.com (F.A.); caterina.foti@uniba.it (C.F.); marco.bellino@policlinico.ba.it (M.B.); 2Section of Molecular Pathology, Department of Precision and Regenerative Medicine and Ionian Area (DiMePRe-J), University of Bari “Aldo Moro”, 70124 Bari, Italy

**Keywords:** lichen amyloidosis, atopic dermatitis, histopathology, dupilumab

## Abstract

Lichen amyloidosis (LA) is a type of cutaneous amyloidosis characterized by brownish hyperkeratotic and itchy papules on the lower leg, back, forearm, or thigh. It is associated with itching and atopic dermatitis (AD) according to an etiopathogenetic mechanism that has not yet been fully elucidated. Currently, the available therapies for this condition include oral antihistamines, laser, cyclosporine, topical corticosteroids, and phototherapy, but, in light of the overlap with AD, Dupilumab may also be indicated. We report the case of a female, 52 years old, who had been suffering from AD and LA for about 27 years. She had lesions attributable to both diseases on the trunk and lower limbs associated with severe itching and had proved resistant to cyclosporine therapy. It was decided to opt for Dupilumab with the induction of 2 fl of 300 mg and maintenance with 1 fl every other week. The therapy proved to be effective, returning a total resolution of both diseases one year after the beginning of the treatment. Dupilumab demonstrated efficacy and safety in the LA related to AD and led to clinical and quality of life improvements in this patient. Therefore, Dupilumab should be considered when treating LA. Further studies should be conducted focusing on the efficacy of the drug on LA (whether or not related to AD), changes in the skin lesions after discontinuation, and the safety of long-term application.

## 1. Introduction

Lichen amyloidosis (LA) is the most common form of primary localized cutaneous amyloidosis (PLCA) [1]. It is characterized by an extracellular amyloid deposition exclusively in the skin [2,3,4,5]. Primary localized cutaneous amyloidosis is divided into four subtypes, namely lichen amyloidosis, macular amyloidosis (MA), biphasic amyloidosis (BA), and nodular amyloidosis (NA).

Clinically, LA presents with clusters of multiple brownish hyperkeratotic papules coalescing in plaques. They may be localized or generalized and are typically distributed on the lower leg, back, forearm, or thigh [2,3,4,5]. At the onset, the lesions are usually unilateral, but a bilateral symmetric distribution pattern can develop over time. The lesions are often associated with itching but may also be asymptomatic [6].

LA is more common in women and among people from Asia and South America than in Europe or North America [1,2].

It can occur as an isolated finding or in association with other diseases, including atopic dermatitis (AD), mycosis fungoides, lichen planus (LP), HIV, multiple endocrine neoplasia type 2 (MEN2A), angiolymphoid hyperplasia with eosinophilia, ankylosing spondylitis, autoimmune thyroiditis, hyperthyroidism, and connective tissue disorders [7].

In light of the clinical presentation and the numerous pathologies with which it can be associated, the diagnosis of LA is not always straightforward.

It undergoes differential diagnoses with keratosis pilare (KP), prurigo nodularis (PN) [8], xanthomas, perforating dermatoses (collagenoses), lichen planus, mycosis fungoides [2], papular mucinosis, myxedema, stasis dermatitis [9], lichen simplex chronicus (LSC), and pretibial pruritic papular dermatitis [6].

A skin biopsy of the affected areas must be performed to obtain a definitive diagnosis, in addition to the analysis of the clinical aspects just mentioned. When the case requires, obtaining a sample of the rectal mucosa or periumbilical fat is recommended to exclude systemic involvement [1].

Different types of PLCA are frequently recognized by common histological features such as PAS positivity, affinity for thioflavin T and Congo Red, and metachromasia following staining with crystal violet or methyl violet. When examined under polarized light, amyloid exhibits distinctive apple green birefringence, also known as dichroism, due to the use of Congo Red. Frequently, amyloid is located in the lichenoid (papular) and macular patterns in the papillary dermis, beneath the dermo-epidermal junction; the acanthosis and hyperkeratosis of the underlying epidermis are frequently observed, and neither macular nor lichenoid amyloidoses involve blood vessels [10,11]. A loose network of unbranched filaments with a diameter of 7.5–10 nm is visible under electron microscopy. The fibrillary anti-parallel beta-sheet structure of the various amyloid precipitates is what unites them ultrastructurally. Protofilaments make up the filaments, and the filaments group together to form fibrils. The fibrils are present in the extracellular space, fibroblasts phagocytose minute amounts of them, and a degeneration of keratinocytes known as filamentous and pyknotic degeneration has been documented in the literature. It is possible that fibroblasts play a direct role in the production of amyloid; however, particular studies on this matter are currently lacking. Unlike the corresponding amyloid protein, the amyloid P component exhibits a pentagonal ultrastructure and is non-fibrillary [12].

Abdominal fat pad fine-needle aspiration biopsy has emerged as the gold standard for diagnosing systemic amyloidosis in recent years, owing in part to its excellent sensitivity and simplicity in smear preparation.

The current therapeutic approach involves topical therapies in the first line aimed at extinguishing the probable underlying inflammatory process and itching where present. The topical drugs recommended to date are essentially topical corticosteroids or topical tacrolimus [3]; dimethyl sulfoxide (DMSO) has been used for several patients but presents some non-negligible side effects (contact urticaria, desquamation of the skin, and a burning sensation), while topical calcipotriol has not returned better effects than corticosteroids [1,3].

If the lesions persist, whether it is a localized LA, it is possible to substitute or add topical drugs to surgical removal (electrodesiccation, a special technique called ‘scraping with the scalpel’, and dermabrasion) or laser removal [13,14,15]. If it is diffuse LA, on the other hand, phototherapy (narrow-band UVB and photochemotherapy with PUVA) and/or systemic drug treatment (oral antihistamines, DMSO, acitretin or other retinoids, cyclophosphamide, and cyclosporine) can be used [3,16,17,18]. In addition, satisfactory results have also been reported using amitriptyline and transcutaneous nerve stimulation [3]. Finally, there are a small number of case reports in the literature in which biological drugs such as Upadacitinib [19], Baricitinib [20], and Dupilumab [7,21,22,23] have been used.

LA significantly impacts patients’ lives because of the blemish it generates and, when symptomatic, because of all the itch-related problems [3]. It is a relatively rare condition, which is the reason why the therapeutic strategies reported in the literature are the result of minor studies and not major clinical trials. For this reason, it is difficult to work out a well-defined treatment protocol to achieve complete remission. In particular, the choice of treatment following the failure of topical drugs and a systemic immunosuppressor is still a tricky issue.

Biological drugs have returned promising results in this regard. However, as explained above, they have still only been used in a small number of cases, and their potential application is an aspect certainly worthy of further investigation.

In this regard, we report the case of a patient with LA associated with AD resistant to conventional therapies who responded to treatment with Dupilumab.

## 2. Case Presentation

A 52-year-old female patient came to our attention on 14 October 2021 with severe itchy symptoms associated with various types of lesions. On the back and upper limbs (particularly at the level of the elbow flexures and the volar surface of the forearms), lesions typical of chronic eczema were observed, namely diffuse patches characterized by erythema, lichenification, and scaling (Figure 1). These patches were the most suggestive for a diagnosis of AD. Rather, plaques with brownish, itchy papules typical of LA were predominant in the lower limbs (Figure 2); there were also some similar, but less infiltrated and less widespread, small patches in the abdominal region. Lastly, diffuse skin xerosis and numerous scratching lesions were observed.

She reported a negative family history of AD and other dermatological conditions.

The patient reported that the first manifestations had occurred about 27 years earlier. The first lesions of LA appeared in 1994 as brownish, itchy papules coalescing in patches on the lower limbs. Over the years, there had been an increase in itching as well as an increase in and evolution of the lesions. In the following years, the patient went to various hospitals, where the hypothesis of systemic amyloidosis was rejected and the histological diagnosis of localized LA and AD was made.

The patient was treated for a considerable time with topical corticosteroids with no effectiveness on either the AD or LA lesions, let alone on the itching. Due to the lack of a response to topical therapy, subsequently, she had been started on cycles of cyclosporine treatment for 4 years previously with little benefit. In particular, it led to a reduction in the itching and AD lesions in 2019 without having any effect on the LA lesions; moreover, the effect did not prove to be long-lasting and the patient experienced a flareup after 4 months.

In order to obtain an objective overall assessment of the patient, some scores widely known in the field of AD were used: Eczema Area Severity Index (EASI), pruritus Visual Analogue Scale (VAS), and Dermatology Life Quality Index (DLQI).

EASI is a score ranging from 0 to 72, where higher values express greater severity. The pruritus VAS is a scale where pruritus is expressed with a score ranging from 0 (absence of itching) to 10 (severe itching). The DLQI is a 10-question questionnaire used for different skin diseases that expresses the severity of the effect of the disease on the patient’s quality of life. The score ranges from 0 (no impact on the patient’s life) to 30 (maximum impact on the patient’s life).

During the first visit, the general evaluation showed the following: EASI 22, pruritus VAS 10, and DLQI 19.

A panel of analysis showed an increase in IgE and eosinophils. Prick tests revealed a sensitization to mites and cypress, while patch tests returned negative results, enabling the exclusion of contact dermatitis.

At our clinic, for a diagnosis of certainty regarding LA, an incisional biopsy was performed regarding the typical infiltrated plaques on the lower limbs. The histopathological investigations reported typical LA findings, such as the deposition of amyloid material at the papillary dermis, beneath the dermo-epidermal junction; there were also some areas with inconsistency regarding the melanic pigment and diffuse inflammatory infiltrate at the middle and superficial dermis. The epidermis was not interested in this infiltration and acts as an innocent bystander with acanthosis, hyperkeratosis, and elongation of the epidermal ridges (Figure 3, Figure 4 and Figure 5). With polarized microscopy, it was possible to appreciate the amyloid deposits beneath the dermo-epidermal junction (Figure 6).

We confirmed the diagnosis of LA and AD. Therefore, it was necessary to find an alternative treatment to resolve the skin problem or at least the itching, given that cyclosporine had proved ineffective.

In light of the presence of AD and the significant itchy symptoms, it was decided to start the patient on Dupilumab treatment. Induction was therefore carried out with 2 fl of 300 mg and maintenance with 1 fl every other week starting on 4 March 2022.

The therapy proved to be effective, showing a marked improvement in the 3-month followup with a reduction in itching and the extension of the eczematous patches, as expected. In contrast, the LA lesions of the legs persisted. The patient presented EASI 15, VAS 0, and DLQI 3.

One year after the beginning of the therapy, a total resolution of the pathology was achieved, also with the unexpected disappearance of the amyloid papules and the achievement of EASI 0 and DLQI and a VAS of 0 (Figure 7).

To date, the patient’s quality of life has exponentially improved thanks to the disappearance of the itchy symptoms and skin blemishes related to AD but also, above all, to LA. At the last checkup, in fact, a total remission of the clinical presentation was appreciated, with the persistence of slight residual skin xerosis and barely perceptible hyperpigmentation on the legs. The lower limbs, as well as the rest of the skin area, do not show any noticeable or scratching lesions, and the patient has not experienced any side effects, so she is still on Dupilumab 300 mg sc every other week.

## 3. Discussion

As mentioned in the introduction, LA is an underrepresented condition in general but especially within the Caucasian population. For this reason, no particularly effective treatment procedure has been devised to date [3]. Often, in fact, good clinical control is not achieved, or the itching goes away without the lesions regressing [7]. In this case, Dupilumab led to a complete resolution of both the AD and LA by achieving not only the disappearance of itching but also of the typical lesions. Therefore, based on the reported results, this drug could be introduced in the treatment of LA.

Our case fits into the context of a small number of minor studies and case reports, which, in the absence of publications with a larger number of patients, are extremely valuable and useful for further pathology investigation. While the etiopathogenetic mechanism of AD has been thoroughly investigated, at least in its main features, that of LA has not yet been clarified. To understand the efficacy of Dupilumab, the pathophysiological association between the two diseases must first be investigated.

It is well-known that IL-4, IL-13, and IL-31 in AD patients are associated with sensory nerve sensitization, itching, and the perpetuation of chronic type 2 inflammation. In particular, in AD, there is an alteration of the normal physiology of the skin barrier, which leads to an increase in the transepidermal water loss (TEWL) and the triggering of Th2-mediated cutaneous inflammatory phenomena. As part of this inflammation, the dysregulation of various cytokine patterns occurs, including that of IL-4, IL-13, and IL-31, which interests us. On the one hand, IL-4 and 13 exacerbate changes in the skin barrier, and, on the other hand, they determine the itchy stimulus either directly or by stimulating IL-31 production. The increased expression of this cytokine is, therefore, one of the main mediators of itching in AD [24].

The etiopathogenetic mechanism of LA has not yet been fully elucidated.

According to the most widely accepted hypothesis, the first pathogenic moment is the apoptosis of the basal keratinocytes. During the cellular end program, they release cytokeratins, which are then coated by autoantibodies, phagocytosed by macrophages, and enzymatically degraded into amyloid K, which triggers an itchy stimulus. Amyloid K is derived in most cases from cytokeratin 5, but cytokeratins 1, 10, and 14 [25] have also been found, and, in one study, deposits of ApoE as well [1]. The primum movens of keratinocyte apoptosis has not yet been identified. Still, some possible triggers, such as itching and the dysregulation of cytokine signaling (particularly of IL-13 and 31 and certain elements regulating their expression), have been proposed [2,26,27,28,29,30,31]. Therefore, the cause of this apoptosis would be found in the first instance regarding the mechanical trauma from scratching, which accounts for the strong association with AD and the initiation of a scratch–itch–scratch cycle [1,2,32]. Chronic inflammation (not necessarily related to itching) could play a role in the genesis of LA as well as that of AD. Indeed, both IL-13 and 31 have been found in the lesions of LA [30], and, furthermore, in cases of familial cutaneous amyloidosis, mutations have been identified in the gene encoding two subunits (IL31RA and OSMR β) of the IL 31 receptor [28,29,31]. Given this, the abnormality of the IL-31 signaling pathway may constitute the molecular substrate of both AD and LA. In this sense, these pathologies could represent two aspects, sometimes isolated, sometimes coexisting, of the phenotypic expression of the aforementioned alteration [33]. Comparing the current management of the two diseases, corticosteroids, calcineurin inhibitors, phototherapy, and immunosuppressors are used in both, including cyclosporine; it should be emphasized that the treatment of patients with LA and AD is even more complex as acitretin and other systemic retinoids cannot be used as they would lead to the worsening of AD [33].

Dupilumab fits perfectly into this context because it acts by antagonizing several elements of the signaling just described. It is an all-human monoclonal IgG4 antibody that binds to the shared alpha subunit of both type I (IL-4Rα/γc) and type II (IL-4Rα/IL-13Rα) IL-4 receptors, inhibiting IL-4 and 13 signal transduction. The interruption of this signaling leads in turn to the blocking of other elements, the most prominent of which is IL-31. Therefore, it acts against the altered expression of IL-4, IL-13, and IL-31, which is responsible for the dysregulation of the type 2 inflammatory reaction [33]. Dupilumab is currently used for treating AD, and its efficacy and good safety profile have been amply demonstrated. It causes an early reduction in itching and AD lesions by acting on several levels [30]. Firstly, it directly blocks the action of IL-4 and IL-13 on sensory neurons. In addition, IL-4 stimulates Th2-cells to produce IL-31, so Dupilumab, by antagonizing the action of IL-4, indirectly reduces the action of IL-31. Dupilumab has also received Food and Drug Administration (FDA) approval for use in PN, and there are several off-label studies concerning its application in additional dermatological disorders [33]. Concerning LA, there are only seven other cases of LA and AD effectively treated with Dupilumab in the literature [21,22,23,24] and one case of isolated generalized LA [7]. In these cases, Dupilumab led to the total disappearance of pruritus after the first 3 months of treatment and to the remission of the lesions between the first 3 and 6 months of treatment.

Comparing our results with those already found in the literature, two aspects are particularly interesting. The first is that the improvement in LA lesions did not occur simultaneously with the improvement in itching and AD but rather with a longer timeframe. Unlike the other cases in the literature, the LA plaques were still present after the first 3 months of treatment, and it took the patient a year to achieve the total remission of the disease. This observation not only raises further questions about the mechanism of action of Dupilumab in the treatment of LA but is an indication that the efficacy of the treatment may not be visible in the first months, as is the case for itching and AD lesions. Additionally, it is interesting to note that ours is the first case to report followup data one year after the start of Dupilumab therapy. This last finding reflects not only the efficacy of Dupilumab but also the absence of relapses obtained with continued therapy.

In the meantime, the mechanism by which Dupilumab leads to an improvement in LA remains unclear. In light of the above, it seems that interrupting the vicious cycle of itchy scratching, but also its direct and indirect effects on the cytokines involved in the Th2 inflammatory response and consequently in the possible pathophysiology of this disorder, are primarily involved.

This thinking is applicable to the use of any biologic drug in the setting of LA, including the two cases in the literature in which anti-jak drugs were used.

In light of the above, LA patients who are unresponsive to topical drugs and systemic immunosuppressor drugs should be directed toward biologic therapy. There is not yet enough information in the literature to choose an anti-jak over Dupilumab or the other way around. However, it should be noted that more cases are reported in the literature in which Dupilumab has been effective. One factor that might enable one to be used rather than the other could certainly be the patient’s comorbidities.

## 4. Conclusions

LA is a condition that significantly impacts the quality of life of the affected patients. It is a relatively uncommon disorder. Hence, several fundamental aspects of this pathology, such as its pathophysiology, diagnostic investigation, and, consequently, therapeutic approach, still need to be fully acknowledged.

Our case report is interesting because it offers insights into aspects of LA that have not yet been fully investigated.

Chronic scratching (whether or not related to AD) could be confirmed as one of the main pathogenetic factors of LA. Patients with abnormalities in IL-31, 13, and 4 signaling might present with coexisting LA and AD phenotypes.

Dupilumab in dermatology already has an indication for AD and PN and has been used in an off-label capacity for numerous other clinical conditions. In light of its effect on several inflammatory cytokines and its usefulness against pruritus, it could be an effective new therapy in patients with DA-related LA and probably in patients with isolated LA.

To confirm this, it proved to be a drug with an excellent safety profile and a good clinical response in our case as well.

The application of Dupilumab in patients with LA is certainly a scientifically relevant topic. Further studies should be conducted focusing on the changes in skin lesions after discontinuing the drug and the efficacy and safety of long-term application.

## Figures and Tables

**Figure 1 diseases-12-00094-f001:**
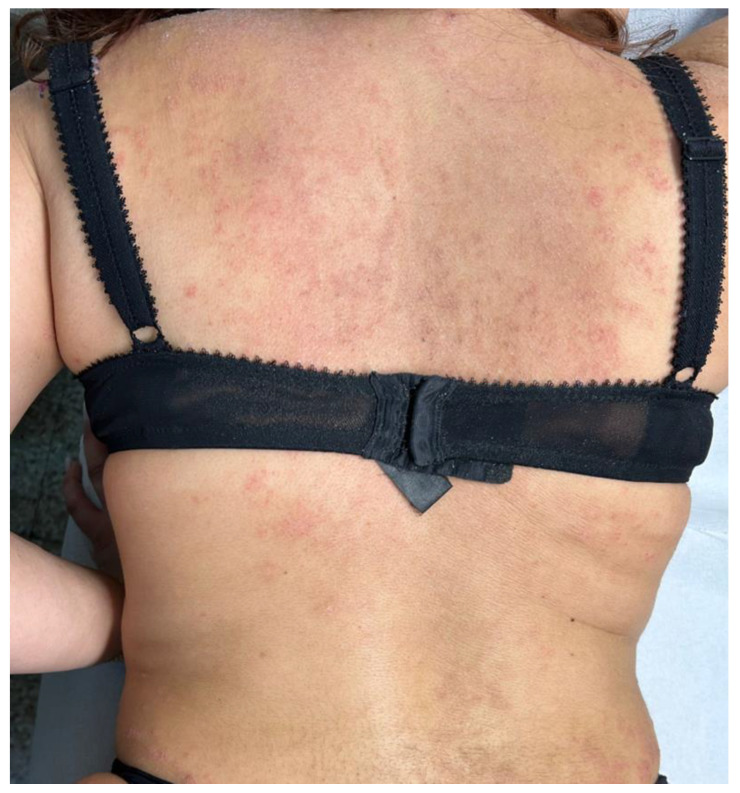
Clinical photograph of AD lesions on the back of the patient.

**Figure 2 diseases-12-00094-f002:**
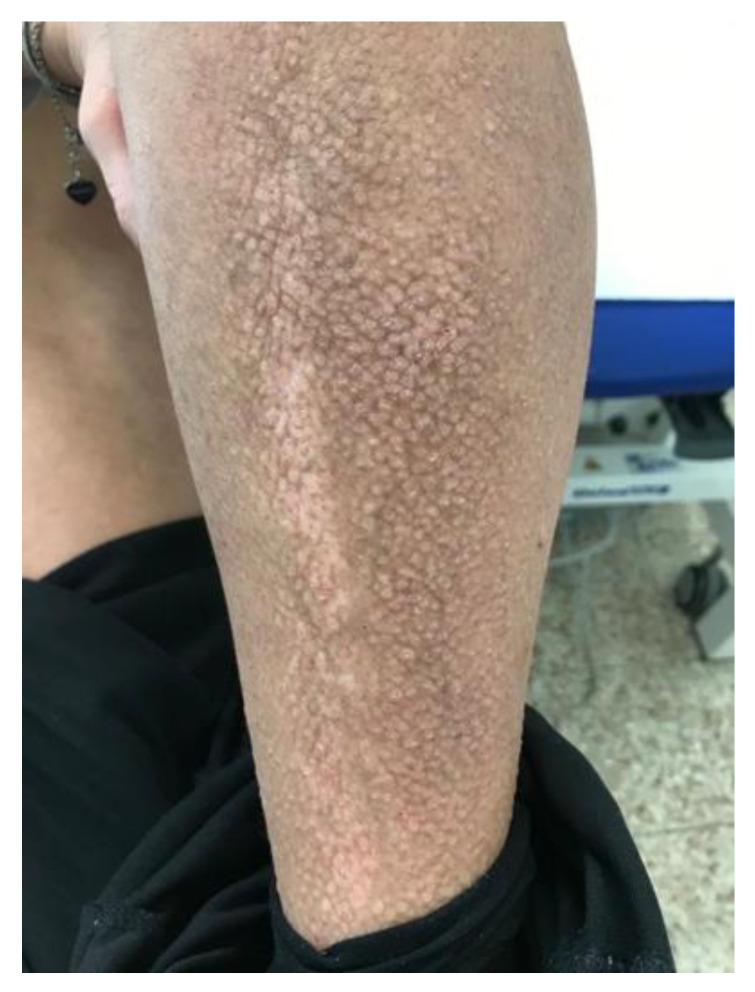
Clinical photograph of lichen amyloidosis before the treatment with Dupilumab. Leg of the patient revealing generalized rippled, dyschromic, brown, and thin plaques.

**Figure 3 diseases-12-00094-f003:**
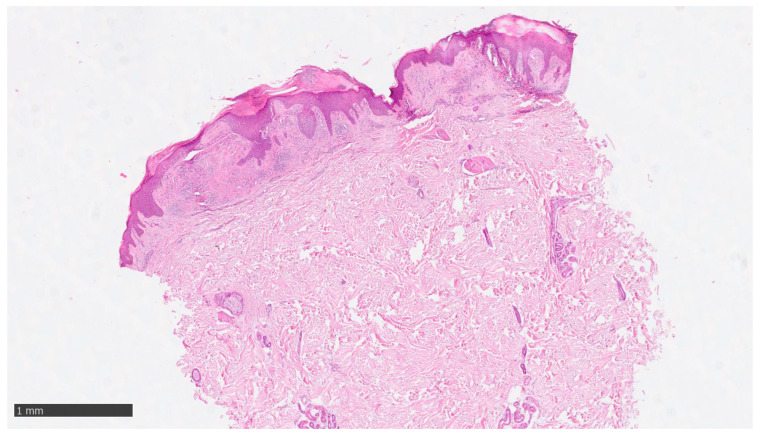
Histological photomicrograph showing an area with hyperkeratosis, some degree of acanthosis, and amyloid deposition in the papillary dermis (hematoxylin–eosin, original magnification 4×).

**Figure 4 diseases-12-00094-f004:**
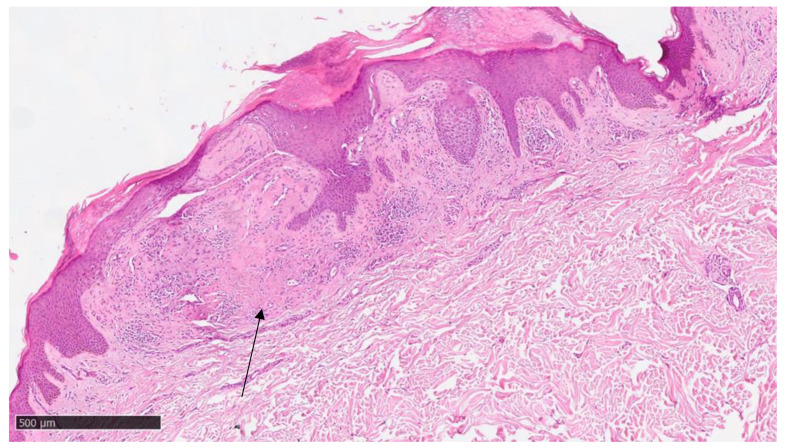
Scanning magnification of the previous picture showing the amyloid deposition beneath the dermo-epidermal junction (black arrow) with the typical cleft (white spaces) and moderate lymphocytic infiltration in the middle dermis (hematoxylin–eosin; original magnification 10×).

**Figure 5 diseases-12-00094-f005:**
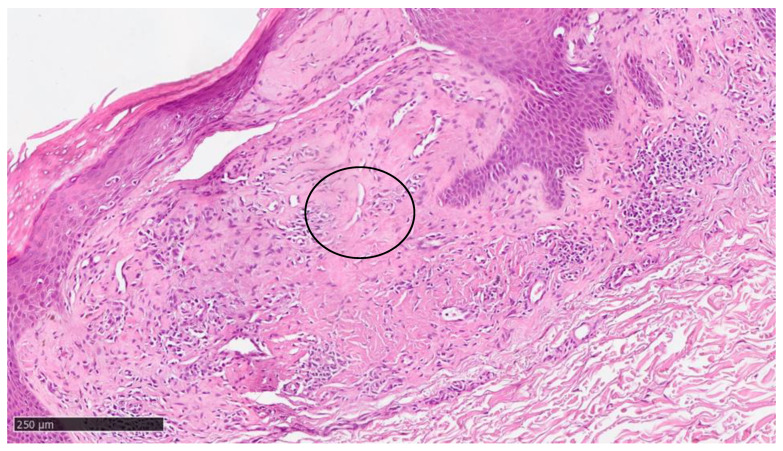
Higher magnification showing some incontinence of melanin within amyloid deposition and with cleft (black circle) (hematoxylin–eosin; original magnification 20×).

**Figure 6 diseases-12-00094-f006:**
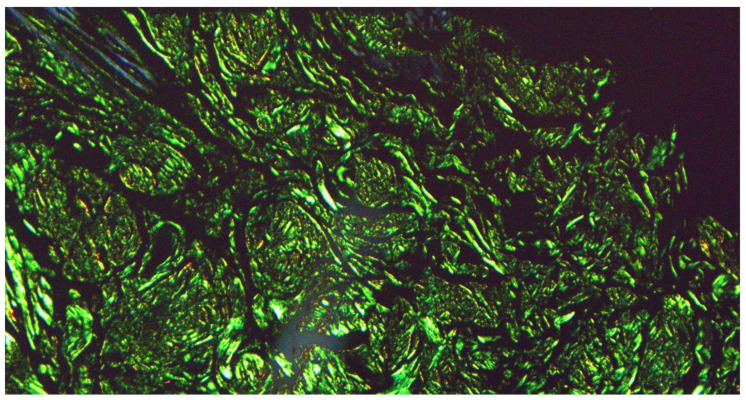
Histological photomicrograph showing characteristic apple green birefringence at polarized microscopy (preparation for Congo Red; original magnification 20×).

**Figure 7 diseases-12-00094-f007:**
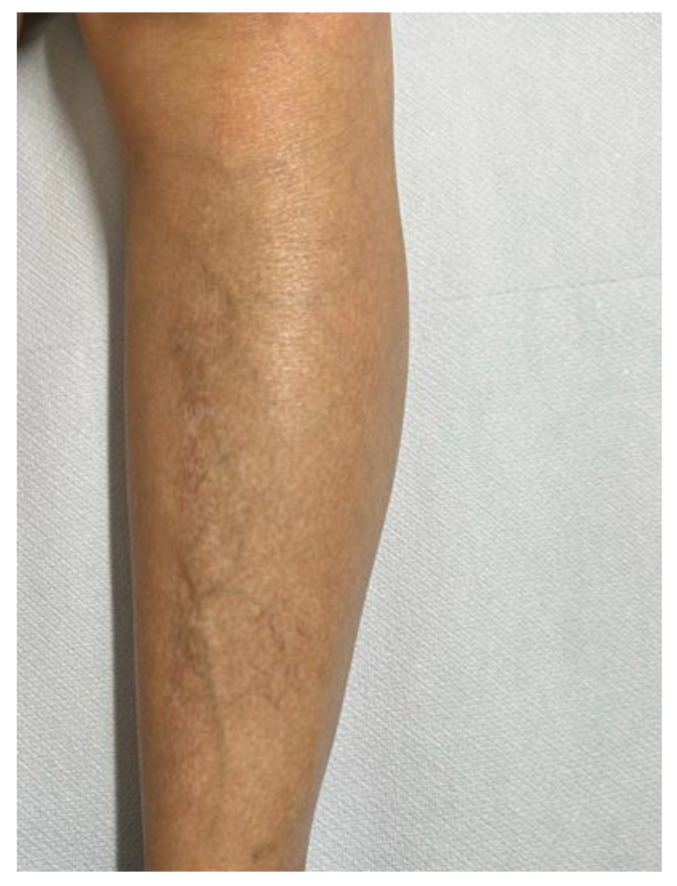
Condition of the patient after one year of Dupilumab therapy showing skin smooth and free of lesions with mild dyschromic aftermath.

## Data Availability

The original contributions presented in the study are included in the article.

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
