# Peer review of "Lichen Amyloidosis in an Atopic Patient Treated with Dupilumab: A New Therapeutic Option"

_diseases, 2024, doi:10.3390/diseases12050094_

Round 1

Reviewer 1 Report

Comments and Suggestions for Authors

Comments on the Quality of English Language

Can be improved

Author Response

REVIEWER 1

1.The introduction part is too long. Please rewrite it so the information is more straightforward and in accordance with the novel findings of the case study.

I tried to shorten the introduction and to make it smoother as suggested.

It is difficult to work out a well-defined treatment protocol to achieve complete remission of LA. In particular, the choice of treatment following the failure of topical drugs and a systemic immunosuppressor is still a tricky issue.  Biological drugs have returned promising results in this regard. We want to make our contribution to the literature by describing our case of good response to treatment with dupilumab. However it has been used in a still small number of cases and its potential application is an aspect certainly worthy of further investigation.

  1. The patient also had AD lesions. Please explain the AD lesions and show the picture of AD as well.

The patient presented typical lesions of chronic eczema on the back and upper limbs (particularly at the level of the elbow flexures and the volar surface of forearms). We were observed diffuse patches characterised by erythema, lichenification and scaling.

We added a picture of the back of the patient to show the lesions of AD.

  1. In the case presentation, the authors only mentioned history of cyclosporin treatment. Was it for the LA or AD treatment or both? Please state here if there were other medications which have been used and failed to give resolution, either for LA or AD. These will support the needed of using Dupilumab for this patient.

The patient was treated for a considerable time with topical corticosteroids with no effectiveness on either AD or LA lesions, let alone on itching. Later she had been started on cycles of cyclosporine treatment for the AD for 4 years previously with little benefit. In particular, it led to a reduction in itching and AD lesions in 2019 without having any effect on LA lesions. Moreover, the effect did not prove to be long-lasting and the patient experienced a flare-up after 4 months.

  1. How did the AD being diagnosed?

It was diagnosed with an incisional biopsy of typical lesions performed at another hospital.

  1. Figure 1: please take a more representative picture for the anatomical point of view. If you want to show the lower leg, then the genu or ankle should be shown first, then you can put a bigger magnification picture to show the detailed skin lesion.

Sorry, it is impossible to find more photos of the patient's condition before the beginning of dupilumab therapy because they are not performed.

  1. Figure 3-5: please put arrows for the characteristic findings of histopathology picture.

Thanks for the suggestion we put the arrows.

  1. I can only find 4 figures, but there are 5 figure legends. Where is the no 5 figure?

We insert the image.

  1. Discussion: …type 2 inflammation…

Thanks for the suggestion we changed the number

  1. Discussion: The aetiopathogenetic mechanism of LA has not yet been fully elucidated → this sentence should go to the next paragraph as they are both part of the mechanism of LA.

Thanks for the suggestion.

  1. Discussion: It is an all-human monoclonal IgG4 antibody that binds to the shared alpha

subunit of both type I (IL-4Rα/γc) and type II (IL-4Rα/IL-13Rα) IL-4 receptor, inhibiting IL-4 and 13 signal transduction and with them other elements, such as IL-31, that are responsible for the dysregulation of the type II inflammatory reaction [33] → this sentence is confusing.

Thanks for the suggestion. I changed the phrase

  1. Discussion page 7: a bracket is need for ref no.33.

I've added a bracket.

  1. Concerning LA, there are only 7 other cases of LA and AD effectively treated with Dupilumab in the literature [21-24] and one case of isolated generalized LA [7]
  2. Do the other reports give similar result as yours?

In these cases, dupilumab led to the total disappearance of pruritus after the first 3 months of treatment and to remission of the lesions between the first 3 and 6 months of treatment. In our case, dupilumab returned similar results but in a longer time frame

  1. Do you have any novel findings in your case which can give new information of dupilumab in concurrent LA and AD patient? I think this is the most important thing hat you need to show to the editor, reviewers, and readers.

The improvement in LA lesions did not occur simultaneously with the improvement in itching and AD, but rather with a longer timeframe. Unlike the other cases in the literature, the LA plaques were still present after the first 3 months of treatment and it took the patient a year to reach total remission of the disease.  This is an indication that the efficacy of the treatment may not be visible in the first months, as is the case for itching and AD lesions. Additionally, it is interesting to note that our case is the first case to report follow-up data one year after the start of dupilumab therapy. This last finding reflects not only the efficacy of Dupilumab, but also the absence of relapse obtained with continued therapy.

Reviewer 2 Report

Comments and Suggestions for Authors

The authors demonstrated a rather typical case of AD and lichen amyloidosis. However, successful treatment with dupilumab makes the case description quite interesting as the treatment options are very limited for lichen amyloidosis. I have, however, some comments to the authors:

1. The introduction is quite long and could be shortened. 

2. When you mention some instruments, they should be shortly explained or references (e.g. VAS, EASI, DLQI)

3. The authors have to provide some stainings to demonstrate deposits of amylond on histology (e.g. Kongo red)

Comments on the Quality of English Language

None

Author Response

REVIEWER 2

  1. The introduction is quite long and could be shortened.

I tried to shorten the introduction and to make it smoother as suggested.

  1. When you mention some instruments, they should be shortly explained or references (e.g. VAS, EASI, DLQI)

I added in the text more information about the different scores mentioned.

EASI is a score ranging from 0 to 72, where higher values express greater severity. The pruritus VAS is a scale where pruritus is expressed in a score ranging from 0 (absence of itching) to 10 (severe itching). The DLQI is a 10-question questionnaire used for different skin diseases that expresses the severity of the effect of the disease on the patient's quality of life. The score ranges from 0 (no impact on the patient's life) to 30 (maximum impact on the patient's life).

  1. The authors have to provide some stainings to demonstrate deposits of amylond on histology (e.g. Kongo red)

Thank you very much. Unfortunately we don't have the possibility to make a picture with polarized light microscopy but we think that the histological features of the amyloid are quite characteristics. sorry.

Reviewer 3 Report

Comments and Suggestions for Authors

The authors submitted a case report of a patient with atopic dermatitis and lichen amyloidosis treated with dupilumab. Since this treatment combination has not been reported yet and given the need for effective and new treatment strategies of lichen amyloidosis, the authors report on a very relevant topic.

The introduction provides sufficient background information on lichen amyloidosis and common differential diagnosis. The references are appropriate. The presented results and the discussion part are reported in a clear and interpretable manner.

Since the mentioned case reports using JAK-inhibitors also showed good results in the treatment of Lichen amyloidosis I would recommend to point out, in which situation/patient constellation you would prefer dupilumab and in which situation JAK inhibitor (in view of personalized medicine).

Furthermore, figure 3 is missing, in the manuscript only the figure caption without figure is visible."

Since this manuscript handles about a case report there is no comment on specific improvement in methodology or specific experiments suitable.

Author Response

REVIEWER 3

The authors submitted a case report of a patient with atopic dermatitis and lichen amyloidosis treated with dupilumab. Since this treatment combination has not been reported yet and given the need for effective and new treatment strategies of lichen amyloidosis, the authors report on a very relevant topic.

The introduction provides sufficient background information on lichen amyloidosis and common differential diagnosis. The references are appropriate. The presented results and the discussion part are reported in a clear and interpretable manner.

Since the mentioned case reports using JAK-inhibitors also showed good results in the treatment of Lichen amyloidosis I would recommend to point out, in which situation/patient constellation you would prefer dupilumab and in which situation JAK inhibitor (in view of personalized medicine).

The LA patient who is unresponsive to topical drugs and systemic immunospressor drugs should be directed toward biologic therapy. There is not yet enough information in the literature to choose an anti-jak over dupilumab or the other way around. However, it should be noted that more cases are reported in the literature in which dupilumab has been effective. One factor that might allow one to be used rather than the other could certainly be the patient's comorbidities.

Furthermore, figure 3 is missing, in the manuscript only the figure caption without figure is visible."

Sorry for the inconvenience.

Since this manuscript handles about a case report there is no comment on specific improvement in methodology or specific experiments suitable.

Round 2

Reviewer 2 Report

Comments and Suggestions for Authors

The authors improved their manuscript, however, they have to provide the proof, that the skin contains amyloid.

Author Response

Dear Reviewer n'2,

thank you very much. We have added a picture of polarized light microscopy showing Amyloid deposits.

A warm greeting.

Round 3

Reviewer 2 Report

Comments and Suggestions for Authors

I do not have further comments.